# Enhancement and Restoration of Scratched Murals Based on Hyperspectral Imaging—A Case Study of Murals in the Baoguang Hall of Qutan Temple, Qinghai, China

**DOI:** 10.3390/s22249780

**Published:** 2022-12-13

**Authors:** Pengyu Sun, Miaole Hou, Shuqiang Lyu, Wanfu Wang, Shuyang Li, Jincheng Mao, Songnian Li

**Affiliations:** 1School of Geomatics and Urban Spatial Informatics, Beijing University of Civil Engineering and Architecture, No. 15 Yongyuan Road, Beijing 102616, China; 2Beijing Key Laboratory for Architectural Heritage Fine Reconstruction & Health Monitoring, No. 15 Yongyuan Road, Beijing 102616, China; 3The Dunhuang Academy, Dunhuang 736200, China; 4Department of Civil Engineering, Toronto Metropolitan University, 350 Victoria Street, Toronto, ON M5B 2K3, Canada

**Keywords:** murals, scratches, enhancement, restoration, principal component transformation, 2D gamma function, triplet domain translation network pretrained model

## Abstract

Environmental changes and human activities have caused serious degradation of murals around the world. Scratches are one of the most common issues in these damaged murals. We propose a new method for virtually enhancing and removing scratches from murals; which can provide an auxiliary reference and support for actual restoration. First, principal component analysis (PCA) was performed on the hyperspectral data of a mural after reflectance correction, and high-pass filtering was performed on the selected first principal component image. Principal component fusion was used to replace the original first principal component with a high-pass filtered first principal component image, which was then inverse PCA transformed with the other original principal component images to obtain an enhanced hyperspectral image. The linear information in the mural was therefore enhanced, and the differences between the scratches and background improved. Second, the enhanced hyperspectral image of the mural was synthesized as a true colour image and converted to the HSV colour space. The light brightness component of the image was estimated using the multi-scale Gaussian function and corrected with a 2D gamma function, thus solving the problem of localised darkness in the murals. Finally, the enhanced mural images were applied as input to the triplet domain translation network pretrained model. The local branches in the translation network perform overall noise smoothing and colour recovery of the mural, while the partial nonlocal block is used to extract the information from the scratches. The mapping process was learned in the hidden space for virtual removal of the scratches. In addition, we added a Butterworth high-pass filter at the end of the network to generate the final restoration result of the mural with a clearer visual effect and richer high-frequency information. We verified and validated these methods for murals in the Baoguang Hall of Qutan Temple. The results show that the proposed method outperforms the restoration results of the total variation (TV) model, curvature-driven diffusion (CDD) model, and Criminisi algorithm. Moreover, the proposed combined method produces better recovery results and improves the visual richness, readability, and artistic expression of the murals compared with direct recovery using a triple domain translation network.

## 1. Introduction

Murals are a precious part of the world’s cultural heritage and have enormous historical and research value. They are the spiritual home of modern man and a symbol of world civilisation, reflecting the social, political, economic, religious, cultural, and artistic development of countries around the world [1]. However, they are not long-lasting and only a few exist as they have been subjected to long-term natural erosion and man-made deterioration, in addition to a slew of other issues. Scratches have emerged on several murals, significantly reducing their aesthetic value and appreciation.

Virtual restoration has attracted research attention in the field of cultural heritage conservation in recent years because of advancements in computer vision technology. New technologies such as computer image processing, graphics, virtual reality, and hyperspectroscopy are gradually being applied to the field of cultural relic protection and restoration [2]. For example, Pei et al. [3] proposed a virtual restoration algorithm for ancient paintings, based on colour contrast enhancement, missing texture synthesis, and the Markov random field model to repair the stains and cracks in ancient paintings and murals. Baatz et al. [4] proposed a binary image restoration method based on the Cahn–Hilliard equation to restore the binary structure of paintings and derived a general grey-scale image restoration method to repair the paintings. Cornelis et al. [5] extracted information on cracks from oil paintings using three methods: filters, top-hat transform, and K-SVD; these methods improved pre-existing patch-based repair techniques to eliminate the detected cracks. Hou et al. [6] proposed a new virtual restoration method for stains based on the maximum noise fraction (MNF) transformation with hyperspectral imaging. This method can fade or eliminate speckles in the image and restore the style of ancient paintings to a large extent without resulting in large data losses. Purkait et al. [7] proposed a semi-automatic mural restoration system based on coherent texture synthesis and high-frequency enhanced diffusion, which realised the restoration of colour murals in Indian temples. Mol et al. [8] proposed an integrated texture and structure reconstruction technique for ancient wall paintings. The method outperformed other reconstruction techniques in terms of image quality and computational efficiency. Wang et al. [9] used the structural information collected from the guidance of painters and line drawings to study mural image restoration and proposed a structure-guided global and local feature weighting method to repair the murals. Cao et al. [10] proposed a method of restoring sooty murals based on the dark channel a priori and the Retinex hyperspectral imaging technique. This approach can effectively reduce the effects of soot on the frescoes, provide additional details that reveal the original appearances of the frescoes, and improve their visual quality.

With the development of artificial intelligence, deep learning is gradually being applied in the field of digital preservation of cultural heritage. Deepak Pathak et al. [11] first proposed an unsupervised visual feature learning method based on contextual pixel prediction using a neural network (NN) approach, which laid the foundation for many subsequent approaches. Alberto Nogales et al. [12] developed a deep-learning model based on GANs for the automatic digital reconstruction of Greek temples. The method automatically repairs Greek temples based on a rendering of the ruins obtained from the 3D model. Gupta et al. [13] proposed a hybrid model that employs R-CNN-based automatic mask generation and image inpainting with partial convolution and automatic mask update using U-Net architecture. The results show that the proposed method is quite effective in the virtual restoration of digitized artworks. Huang et al. [14] solved the mural degradation detection problem with a multi-path convolutional neural network (CNN) and designed an eight-path CNN. The effectiveness and efficiency of the method were verified by extensive experiments. Wang et al. [15] proposed a Thangka mural restoration method based on multi-scale adaptive partial convolution and stroke-like masks for Tibetan Thangka murals. Li et al. [16] proposed a generative-discriminator network model based on artificial intelligence algorithms for digital image restoration of damaged ancient wall paintings; in adversarial learning. The discriminator network model was optimised in this study, and the proposed algorithm effectively restored wall paintings with point-like damage and complex texture structures.

Scratches are different from small defects such as cracks, as they punctate losses, are often seen in large areas of structural damage, and are irregular in shape. This poses difficulties for the restoration of wall paintings. The majority of existing literature has focused on the restoration of punctate loss, fading, cracks, and other defects in the mural images. However, to date, there are few methods for recovering large areas of scratch damage in mural images. We observed that the scratches in the mural were similar to the creases in the old photograph. The scratches in the mural and the folds in the old photograph are both large, elongated, and white in appearance. In our study, we used pretrained models [17] of old photographs to remove scratches from the murals. However, there are still differences between the murals and the old photographs, and through extensive experiments, we found that direct scratch removal using the triplet domain translation network pretrained model did not produce the best results.

Therefore, we opted to use spectral information for line enhancement, a 2D gamma function to enhance local dark information, and a triplet domain translation network pretrained model and a Butterworth filter to virtually restore scratches. After radiation correction, the mural’s hyperspectral data were subjected to principal component analysis (PCA) and high-pass filtering, producing improved hyperspectral images, and thereby improving the linear information and contrast between scratches and the backdrop of the mural. Second, the mural’s improved hyperspectral picture was synthesised to a true colour image that was converted to the HSV colour space. A multi-scale Gaussian function was used to estimate the image’s lighting component. Thereafter, a 2D gamma function was used to correct the brightness and overcome the problem of poor light in murals. Finally, the triplet domain translation network pretrained model received the augmented mural images. The local branch of the network performs overall image quality restoration to address fading and noise in the image. A partial nonlocal block was used to recover structured defects in the image to resolve scratches, and a Butterworth filter was applied to make the final result clearer. This methodological approach will help us gain a sharper and more comprehensive understanding of murals.

In this study, we have used the murals of the Baoguang Hall at Qutan Temple as an example of virtual restoration of large scratch lesions on their surfaces. The main contributions of this study are summarized as follows.

(1)A method combining linear information enhancement and triplet domain translation network pretrained model is proposed to recover the scratch lesions in the mural images, which includes using hyperspectral data for principal component analysis and enhancing the first principal component with high-pass filtering, replacing the original first principal component with the enhanced first principal component by principal component fusion, and recovering the data dimension with principal component inversion to produce an improved hyperspectral mural image. Then, a triplet domain translation network pretrained model was used to complete the repair of the scratch lesions. In addition, we added a Butterworth high-pass filter after the pretrained model restoration to produce sharper and higher visual quality mural restoration results. As such, this study fills a gap in the existing literature on the virtual restoration of mural scratches.(2)The 2D gamma function light uneven image correction algorithm is applied to the mural to solve the problem of local low luminance, enhance the information in the dark areas, and provide more accurate restoration results.(3)The proposed method can provide auxiliary reference and support for the actual restoration of murals. It is helpful to provide conservators with a scratch-free appearance of the murals before the restoration process begins. In addition, the work in this study is an attempt to provide novel ideas for the digital conservation of wall paintings for World Heritage sites.

The rest of the paper is organized as follows. Section 2 describes the experiments designed for the enhancement and restoration of scratched murals, including the experimental materials and acquisition techniques used and the workflow of the experiments. Section 3 describes our experimental results and visual comparisons of the murals before and after restoration. In Section 4, the results of restoration by omitting one of the steps proposed in this paper are discussed in detail, and the comparison of this method with other existing virtual restoration techniques is presented. Section 5 presents the conclusions of this study.

## 2. Materials and Methods

### 2.1. Materials

#### 2.1.1. Murals

A scratch is a mark produced by an external force that damages mural patterns [18]. Mural patterns are frequently ruined by scratches, lowering their creative value significantly. The mural data used in this study were from the murals on the east wall of Baoguang Hall, Qutan Temple, Ledu District, Haidong City, Qinghai Province, China. According to historical records, Qutan Temple is a Tibetan Buddhist monastery founded in 1392. The large colourful murals in the temple were created by the court painters of the Ming and Qing dynasties, as shown in Figure 1a. Owing to their refined painting techniques and striking ideas, these murals are considered to be some of the best works of art ever produced. However, several of these works have been extensively damaged, their magnificent patterns have gone unfinished, and their aesthetic and decorative values have diminished. The hyperspectral data of the murals were collected and analysed to eliminate the influence of scratches and restore the original appearance. In this study, two experimental areas were selected, as shown in Figure 1b,c. The images were all true colour images synthesized from hyperspectral images based on wavelengths of 460.20, 549.79, and 640.31 nm.

#### 2.1.2. Data Acquisition

Data from the experimental area were collected with the hyperspectral image analysis system THEMIS-VNIR/400H from Themis Vision System, USA, with a spatial resolution of 1392 × 1000 pixels and a sampling interval of 0.6 nm. The spectral resolution was 2.8 nm, and the images were collected in 1040 bands ranging from 377.45 (visible light) to 1033.10 nm (near-infrared). During the data collection process, the distance between the hyperspectral camera and the mural was about 1 m. Two halogen lamps were used as light sources.

### 2.2. Methods

Figure 2 shows the overall framework of the enhancement and restoration method for the scratched murals, which includes four main steps:(1)Data denoising using radiometric correction;(2)Mural line information enhancement based on principal component transformation, high-pass filtering, and principal component fusion;(3)Enhancement of local dark information in the mural using multiscale Gaussian and 2D gamma functions;(4)Extraction and repair of scratched murals using a triplet domain translation pretrained network model and Butterworth high-pass filter.

**Figure 2 sensors-22-09780-f002:**
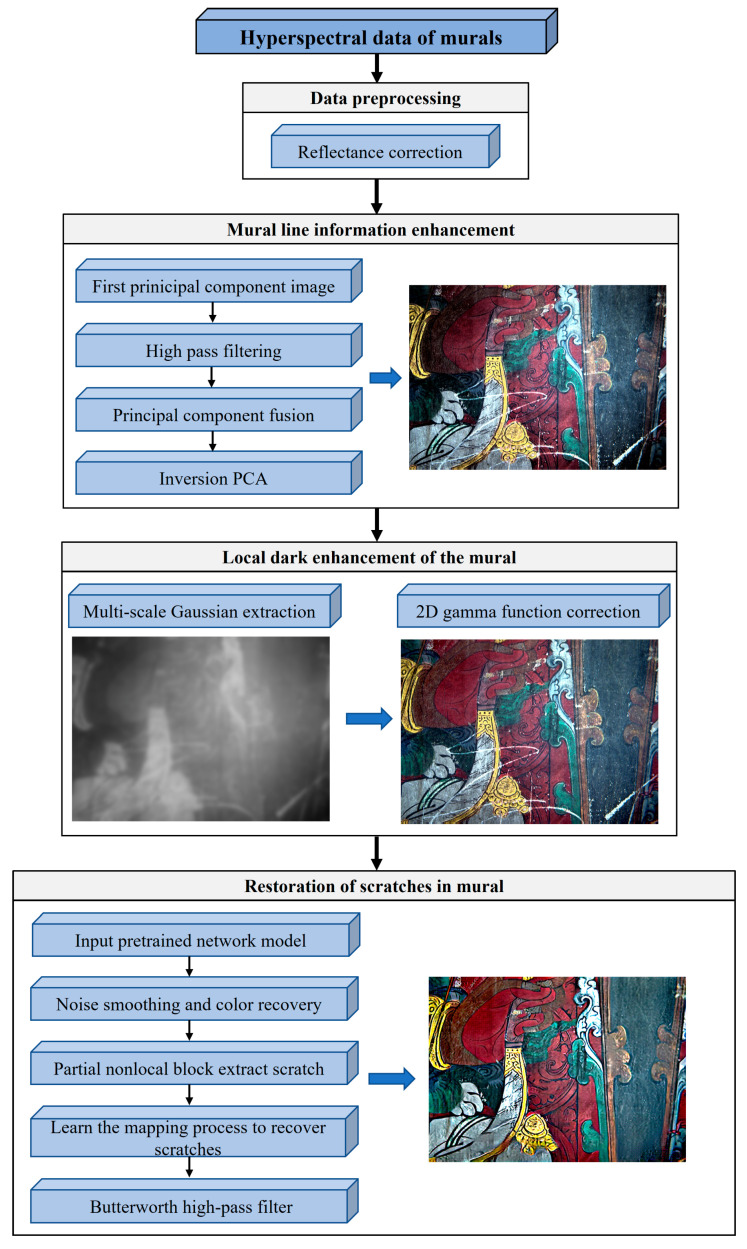
Overall restoration process.

#### 2.2.1. Data Preprocessing

Hyperspectral techniques allow extraction of the maximum amount of information from murals without damaging them owing to their non-contact, non-destructive detection characteristics [19]. Hyperspectral images generally have many bands, a wide spectral range, spectral resolution of the order of nanometres, and a wealth of spectral information. Thus, they can help in the restoration of murals.

During data acquisition with hyperspectral imaging systems, data can be affected by ambient light and the instrument’s dark current noise. Reflectance correction can be used to reduce this type of noise with the following correction formula:(1)$$R=\frac{R_{raw}-R_{dark}}{R_{white}-R_{dark}}\times 99\%,$$
where R is the reflectance, $R_{raw}$ is the collected hyperspectral data, $R_{dark}$ is the dark current data, and $R_{white}$ is the standard reflector data; the reflectance of a standard reflector is 99%.

#### 2.2.2. Line Information Enhancement

We synthesized true colour images in the red, green, and blue bands at wavelengths of 640.31, 549.79, and 460.20 nm, respectively, to meet the memory requirements of the network restoration model and maintain the integrity of the scratched murals, as a direct restoration would be inaccurate and cause the lines in the murals to fade. Before restoring the network, we used the hyperspectral data from the murals to improve the line information and produce better restoration outcomes. A PCA of the hyperspectral images of the murals to be restored was carried out to compress or combine image information from multiple bands into one image [20]; the contribution from the information in each band was maximised in the new image. The first principal component was selected, as it contained most of the information for all of the bands, and high-pass filtering was performed. High-pass filtering is generally used to reduce blur in an image by enhancing the high-frequency components and eliminating the low-frequency components of the image while maintaining the high-frequency information; it is often used to enhance the information of textures and edges [21]. The high frequencies extracted by the default high-pass filtering in the ENVI 5.3 software from Exelis Visual Information Solutions, USA, were added back to the first principal component image of the original mural to obtain a clearer image. High-pass filtering is normally carried out by applying a transform kernel with a high central value, usually surrounded by negative weights. Using a 3 × 3 template, we calculated the following equation:(2)$$H\left(x,y\right)=\left[\begin{array}{ccc} -1 & -1 & -1\\ -1 & 8 & -1\\ -1 & -1 & -1 \end{array}\right],$$
where $H\left(x,y\right)$ is the high-pass filtering convolution template.

Principal component fusion is a PCA transformation of the n-band spectral images to obtain the n principal components based on the vector eigenvalues. The high-resolution panchromatic image is histogram-matched to the first principal component to ensure the grey mean and variance of the panchromatic image agree with those of the first principal component image; the matched panchromatic image is then directly replaced by the first principal component image. Finally, the high-resolution spectral fusion image is obtained by PCA inverse transform processing; this image retains the high-frequency information of the original image. Through this processing, the detailed features of the target are more clearly defined, and spectrally richer images are obtained [22]. The first principal component map after high-pass filtering is used as the panchromatic image to replace the original first principal component image. They are inversely transformed by PCA to recover the dimensionality of the original hyperspectral images with enhanced line information, thus enhancing the identification of scratched areas.

#### 2.2.3. Enhancement of Local Darkness

A 2D gamma function, light inhomogeneity image correction algorithm was applied to the murals to solve the problem of localised low brightness in their images [23]. According to Retinex theory, the brightness component of a real scene is mainly present in the low-frequency part of the image with smooth overall changes, while the reflection component is mainly present in the high-frequency parts of the image such as edges and textures, with more intense changes [24].

As the multiscale Gaussian function could effectively compress the dynamic range and accurately estimate the brightness component of the scene [25,26], it was applied to extract the brightness component of the image before performing luminance correction, with the following mathematical expression:(3)$$G\left(x,y\right)=\lambda exp\left(-\frac{x^{2}+y^{2}}{c^{2}}\right),$$
where c is the scale factor and λ is the normalisation constant; this equation satisfies $\iint G\left(x,y\right)dxdy=1$.

The convolution of the Gaussian function with the original image yielded an estimate of the light component with the following mathematical expression:(4)$$I\left(x,y\right)=\overset{N}{\underset{i=1}{\sum }}\omega_{i\;}\left[F\left(x,y\right)*G_{i\;}\left(x,y\right)\right],$$
where $I\left(x,y\right)$ is the light component value extracted and weighted by Gaussian functions at different scales at the point $\left(x,y\right)$; $F\left(x,y\right)$ is the input image; $G_{i\;}\left(x,y\right)$ is the Gaussian function; * denotes the convolution; $\omega_{i\;}$ denotes the weight; and i=1, 2,…,N is the number of scales used, where N = 1 for single scale and N > 1 for multiple scales.

The 2D gamma function can effectively correct the brightness of an image without changing its overall magnitude [27]. It converts an image from the RGB space to the HSV colour space and changes the brightness of its V luminance component according to the distribution of the image’s light component. This function is used to enhance the light values in dark regions and reduce the light values in bright regions.

Images in HSV space are converted to RGB (red, green, blue) colour space to correct for the brightness of localised darkness in the murals. The mathematical expression is as follows:(5)$$O\left(x,y\right)=255{\left(\frac{F\left(x,y\right)}{255}\right)}^{\gamma },\;\gamma ={\left(\frac{1}{2}\right)}^{\frac{m-I\left(x,y\right)}{m}},$$
where $O\left(x,y\right)$ is the light value of the corrected image, γ is the exponential value used for luminance enhancement, and m is the mean value of the luminance of the light component.

#### 2.2.4. The Pretrained Model and Butterworth High-Pass Filter for Recovery

As described above, the scratches in the mural were more similar to the creases in the old photograph. We therefore applied the pretrained model of the triplet state domain translation network [17] used to restore the old photographs to the mural restoration work. The network is described as follows.

The triplet domain translation network model consists of two variational autoencoders (VAEs) and a mapping network T, each of which can be considered as a separate module. In Figure 3. the mural hyperspectral data are denoted as r, the synthetic data as x, and the truth data corresponding to the synthetic data as y. Of these, the network model synthesized data are broken photos formed by degradation of intact photos and the true value data are intact photos. The real mural hyperspectral data, the synthetic data, and the true value data corresponding to the synthetic data are placed in three different domains; these domains were interconverted in this network model. The real mural’s hyperspectral data domain is denoted as R, the synthetic data domain as X, and the truth data domain corresponding to the synthetic data as Y. The scratches in the mural hyperspectral data are recovered by interconversion and learning between the three domains. $Z_{X}$, $Z_{Y}$, and $Z_{R}$ are the latent spaces corresponding to the mural hyperspectral data, the synthetic data, and the truth data corresponding to the synthetic data, respectively. $E_{R,X}$ and $E_{Y}$ are the encoders and $G_{R,X}$ and $D_{Y}$ are the decoders that form the VAE. The images from the three different domains are mapped to the corresponding hidden spaces by the VAE; the hidden spaces of the mural hyperspectral data and synthetic data are aligned as closely as possible. The recovery of the mural hyperspectral data r is achieved by learning a mapping process from the hidden space $Z_{X}$ of the synthetic data x to the hidden space $Z_{Y}$ of the synthetic data corresponding to the real value data y. A local branch in the network performs global noise removal and colour restoration for non-structural defect problems such as noise and fading. Another branch consists of partial nonlocal block and several residual blocks. It is primarily aimed at structural defects such as scratches. It uses a mask as input to pre-empt pixels in damaged areas of the mural image from being used to repair diseased areas, and this mask prediction network is a U-net.

Here, $VAE_{1}$ consists of an encoder $E_{R,X}$ and a decoder $G_{R,X}$, which encodes the mural r and the synthetic image x into their corresponding hidden spaces $Z_{R}$ and $Z_{X}$, respectively, and then recovers them afterwards, and causes the potential encodings both conform to a Gaussian distribution. $VAE_{2}$ is used to train the true value data y. The expression for the objective function of the mural r to be restored is as follows:(6)$$L_{VAE_{1}}\left(r\right)=ΚL\left(E_{R,X}\left(Z_{r}\left|r\right.\right)N\left(0,I\right)\right)+\alpha E_{Z_{r}\sim E_{R,X}\left(Z_{r}\left|r\right.\right)}\left[∥G_{R,X}\left(r_{R\to R}\left|z_{r}\right.\right)-r{∥}_{1}\right]+L_{VAE_{1},GAN}\left(r\right)\;,\;$$
where the first term is the ΚL regular term to constrain the distribution of the potential encoding to be close to a Gaussian distribution, and $E_{R,X}\left(Z_{r}\left|r\right.\right)$ denotes the prior probability distribution obeyed by $Z_{r}$ obtained through $E_{R,X}$ at input r. The second term denotes the loss between the recovered result by VAE encoding and the input data r, constraining the main information of the image captured by the hidden encoding; the third term is the least squares generative adversarial network loss rate constraining the VAE generated result to be more detailed. As the mural data r share a VAE with the synthetic data x, the inclusion of an adversarial network is used to further approximate the latent space of both, whose loss is defined as
(7)$$L_{VAE_{1,}GAN}^{latent}\left(r,x\right)=E_{x\sim X}[D_{R,X}{\left(E_{R,X}\left(x\right)\right)}^{2}]+E_{x\sim R}[(1-D_{R,X}\left(E_{R,X}\left(r\right)){)}^{2}\right],\;$$

Combined with the latent adversarial loss, the total objective function for $VAE_{1}$ becomes
(8)$$\underset{E_{R,X,}G_{R,X}}{\min}\underset{D_{R,X}}{\max}L_{VAE_{1}}\left(r\right)+L_{VAE_{1}}\left(x\right)+L_{VAE_{1,}GAN}^{latent}\left(r,x\right),\;$$

As the mural data to be recovered and the synthetic data are already well domain aligned in the hidden space, the mapping from the hidden space $Z_{X}$ to the hidden space $Z_{Y}$ learned through the paired data $\left(x,y\right)$ can also be well generalised to the recovered mural. At this stage, the two VAEs are fixed and then the mapping network T of the two hidden spaces is learned. The expression of the loss function of this mapping network T is as follows:(9)$$L_{T}\left(x,y\right)=\lambda_{1}L_{T,\ell_{1}\;}+L_{T,GAN\;}+\lambda_{2}L_{FM\;},$$

In Equation (9), The first item is the latent space loss, $L_{T,\ell_{1}\;}=E∥T$($Z_{x}-Z_{y}$)${∥}_{1}\;$; the second item is the adversarial loss $L_{T,GAN\;}$, to encourage the ultimate translated synthetic image to look real; and the third term is the perceptual loss derived using the VGG network [28].

One of the triplet domain translation network datasets is from the Pascal VOC dataset [29] and the other dataset is a collection of old photographs. The network adopts the Adam solver [30] with $\beta_{1}$ = 0.5 and $\beta_{2}$ = 0.999. The learning rate is set to 0.0002 for the first 100 epochs, with linear decay to zero thereafter. Here, α = 10, $\lambda_{1}$ = 60, and $\lambda_{2}$ = 10 in Equations (6) and (9).

To produce clearer results of the mural restoration, we apply the Butterworth high-pass filter to the restored results of the network to produce visually clearer and sharper restoration results of the mural images. The transfer function of the Butterworth high-pass filter is shown in Equation (10):(10)$$H\left(u,v\right)=1/\left\{1+{\left[D_{0}/D\left(u,v\right)\right]}^{2n}\right\},$$
where $D_{0}\;$is the cut-off frequency, $D\left(u,v\right)$ = $\sqrt{u^{2}+v^{2}}$ is the distance from the point $\left(u,v\right)$ to the origin of the frequency plane, and n is the order of the filter.

## 3. Results

### 3.1. Enhancement of Mural Line Information

As shown in Figure 4a, the mural is badly damaged by scratches. Figure 4b shows the high-pass filtered first principal component image of experimental region 1. As seen in Figure 4c, this image is replaced by the original principal component image using principal component fusion; the remaining principal component images are inverted to obtain a mural image with enhanced line information. Using linear information enhancement prevents the distortion of the mural colours and enhances the information of the lines and details in the background. Observed from the enlarged views Figure 4d–g, the enhanced background black lines are clearer and the colours are more realistic, thus improving the distinction between the scratches and background. These enhancements help the subsequent network pretrained model to identify scratches and improve the problem of ambiguous results owing to direct recovery using the network pretrained model.

### 3.2. Enhancement of Local Darkness Information for Murals 

As shown in Figure 5a,d, the surrounding corners of the murals are dark, and their original fine patterns and colours cannot be seen. Localised darkness in the mural was enhanced before the virtual restoration of the scratches to achieve the best post-restoration visual effect. The light components of the mural were first extracted using a multi-scale Gaussian function with the number of scales i chosen to be 3, where the scale factor c was chosen to be 15, 80, and 250 and the weight factor of the light components extracted at each scale was set to 1/3. The results are shown in Figure 5b,e. Based on the distribution characteristics of the extracted light components, a 2D gamma function was used for correction. The results are shown in Figure 5c,f. Visually, the correction cleared the otherwise unreadable patterns around the perimeter, restoring the dark parts of the mural. This enhances the overall visual impact of the restored mural.

### 3.3. Restoration of Mural Scratches

We use Python to provide the experimental environment needed to build the pretrained model. In this case, the images of the mural to be restored were fed into the pretrained model as a test set. The details of the experimental environment are shown in Table 1.

The enhanced mural images obtained in the previous step were fed into the network model as test sets to repair the scratches. The first step was the full recovery of the unstructured degradation of the murals using local branches. Thereafter, for scratches, the image was segmented using the U-net network [31] in the partial non-local block; the detected scratch points were set to 1 and the others to 0. The mask file was generated and the trained triplet domain translation network restoration model was invoked to restore the image as a whole and process the mask. The repair model was invoked where there were scratches. The scratches were filled by bilinear interpolation and a global wide-area search. Finally, the Butterworth high-pass filter was used to make the resulting high-frequency detail richer and the content clearer in the mural images; the order n = 2, and the cut-off frequency $D_{0}$ = 30 are chosen as parameters for the filter. n = 2 has no significant ringing effect and produces a blurring effect at higher values of n. The higher the cut-off frequency $D_{0}$, the lower the frequency components that are filtered out and the higher the frequency components that are lost. The higher the cut-off frequency $D_{0}$, the more low-frequency components are filtered out, and the more high-frequency components are lost. Therefore, we choose the middle value of 30 as the value of $D_{0}$. Figure 6 shows the scratch extraction and repair results for Area 1 and 2.

### 3.4. Visual Comparison

Figure 7 shows the visual comparison of the scratched murals before and after enhancement and restoration. The results showed that the proposed method enhanced the line information of the mural using principal component transformation and high-pass filtering. It also restored the partial darkness of the mural using optical component extraction and 2D gamma function correction. Further, it achieved the automatic restoration of the scratch damage of the mural by combining the triplet domain translation network model and Butterworth high-pass filter. We successfully restored the pattern information of the scratched murals and provided a reference for the subsequent conservation and restoration of other ancient painted murals.

## 4. Discussion

### 4.1. Combination of Different Steps

To illustrate that the direct use of pretrained models to recover scratch lesions is not ideal, we chose to omit the enhancement of mural line information, local dark enhancement, and Butterworth high-pass filtering steps, while keeping the other processes unchanged to verify the feasibility of the combined enhancement and restoration method proposed in this work. Considering Area 1 as an illustrative example, the results are shown in Figure 8. As can be seen in Figure 8, the omission of the enhancement of mural line information and Butterworth high-pass filtering reduced the accuracy of the network model in detecting scratches and fill effects, resulting in parts of the mural that were not scratched being incorrectly restored. After restoration, the image clarity suffered a loss. Dark enhancement of some of the paintings and Butterworth high-pass filtering has been neglected, leaving their darker areas unclear even after restoration, and the overall viewing of the mural compromised. Similarly, the Butterworth high-pass filter was omitted and some pixels in the restoration result of the mural looked less clear than in the original image. In order to validate the significance of the methods presented in this paper, we introduced the image evaluation metrics of mean gradient, edge strength, and space frequency to evaluate the effectiveness of the restoration process in this paper more objectively. Considering objective quality evaluations, the results in Table 2 show that the average gradient, edge strength, and space frequency of the proposed complete process method were greater than those of the other steps of the truncated process.

### 4.2. Comparison of Scratched Mural Repair Methods

The total variation (TV) model, curvature-driven diffusion (CDD) model, and Criminisi algorithm are all commonly used approaches for image restoration and can be used for the virtual restoration of scratched murals. A comparative analysis of the methods in this study was carried out. The TV model [32] and CDD model [33] are based on partial differential equations. The pixel diffusion principle was mainly used to find the structural information near the scratched area and rely on the shortest straight line to connect this information to achieve image restoration. The Criminisi algorithm is based on priority order sample filling restoration to drive sampling based on the iso-illumination line process and achieve image restoration by searching for the best similar matching blocks for texture replication [34]. In this study, the enhanced and corrected scratched murals were restored by applying the methods described above, and all of the traditional methods used for comparison used the same enhancement steps as the proposed methods, to ensure that this comparison was fair. The subjective visual effects of the different restoration methods for Areas 1 and 2 are shown in Figure 9.

As shown in Figure 9, the TV and CDD models are diffusion models. The information of the intact area is diffused to the area to be repaired, making it easier to produce blurring and leave repair marks when repairing areas with severe scratches. The Criminisi algorithm is better at repairing small defective areas but cannot completely repair areas affected by larger and longer scratches because it uses the concept of fast matching replication of the texture structure. When there are several random missing pixels, it is impossible to form an effective block match, resulting in unsatisfactory repairs. The proposed method gives the most satisfactory results considering the subjective virtual restoration of scratch-damaged murals, which appear better after repair. To better illustrate the subjective quality, we use a scoring method to compare the proposed method with other methods. We collected the subjective opinions of 15 researchers, experts and students from the fields of heritage conservation and image processing. The researchers were asked to rate the results of the different methods according to the quality of the mural restoration. The full score is 10 points, and the restoration results are scored according to the degree of satisfaction, and then the average score is calculated as the final score. The results are shown in Table 3. The results show that the average score of our method is higher than those of other methods, which indicates the obvious advantage of our method.

### 4.3. Applicability of the Proposed Method

To demonstrate the generality of the method in this paper, we selected other wall images with scratches from the Baoguang Hall of Qutan Temple (a1, b1, c1, d1, e1, and f1). In addition, we also restored two scratched murals on the west wall of Guanyin Hall in Heilongmiao Village, using data collected during July 2017 (g1 and h1). The results of their recovery are shown in Figure 10.

## 5. Conclusions

In historical conservation applications, repairing scratched murals is a challenging task. We applied the pretrained model of old photographs to the restoration of fresco scratches. However, there are some differences between mural images and old photographs, and the direct use of this model for mural restoration could cause some of the restoration results to be overly smoothed, resulting in slight blurring or even incorrect restoration of some areas. In response, we proposed a combination of enhancement and restoration that goes some way to improving the outcome of the mural restoration. Linear information enhancement was used to improve the discrimination between scratches and the background in murals. Furthermore, a combination of optical component extraction and 2D gamma function correction was used to enhance local dark information in the mural. The triplet domain translation network pretrained model and Butterworth high-pass filter were used to restore the scratches on the murals. The results produced good aesthetic outcomes, and the repair of scratched murals using different methods was evaluated objectively. In addition, because of the non-reproducible nature of murals as cultural artefacts, scratched murals do not have their intact counterparts as real values to use as a reference. This will be considered in our further work to create some mockups of the murals as true values and artificially carve some scratches on them as synthetic murals. In this way, enough mural images can be collected to create a mural dataset, and a network model can be trained specifically for mural recovery, which helps develop more appropriate restoration techniques to produce better virtual restorations of scratched murals.

## Figures and Tables

**Figure 1 sensors-22-09780-f001:**
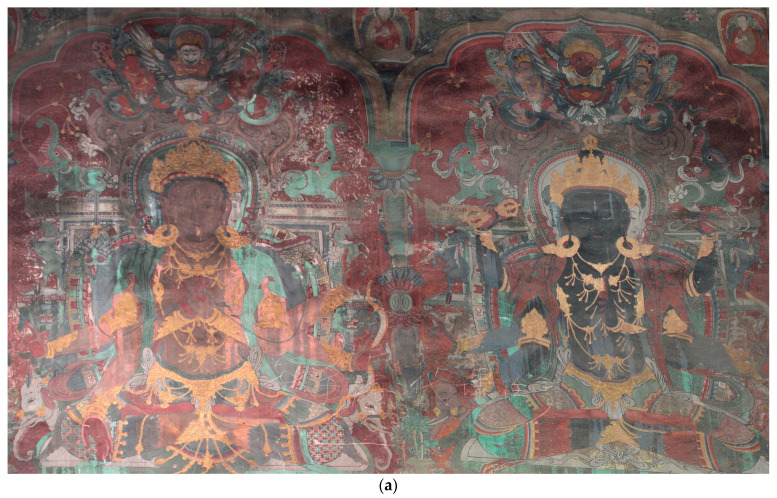
Mural images and two scratch study areas: (**a**) Buddha 2 and 3 from the southeast wall of the Baoguang Hall at Qutan Temple, (**b**) image of the first study area, and (**c**) image of the second study area.

**Figure 3 sensors-22-09780-f003:**
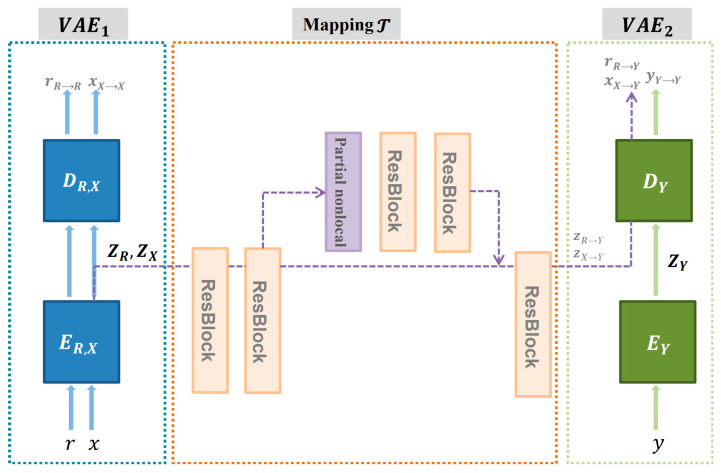
Triplet domain translation network.

**Figure 4 sensors-22-09780-f004:**
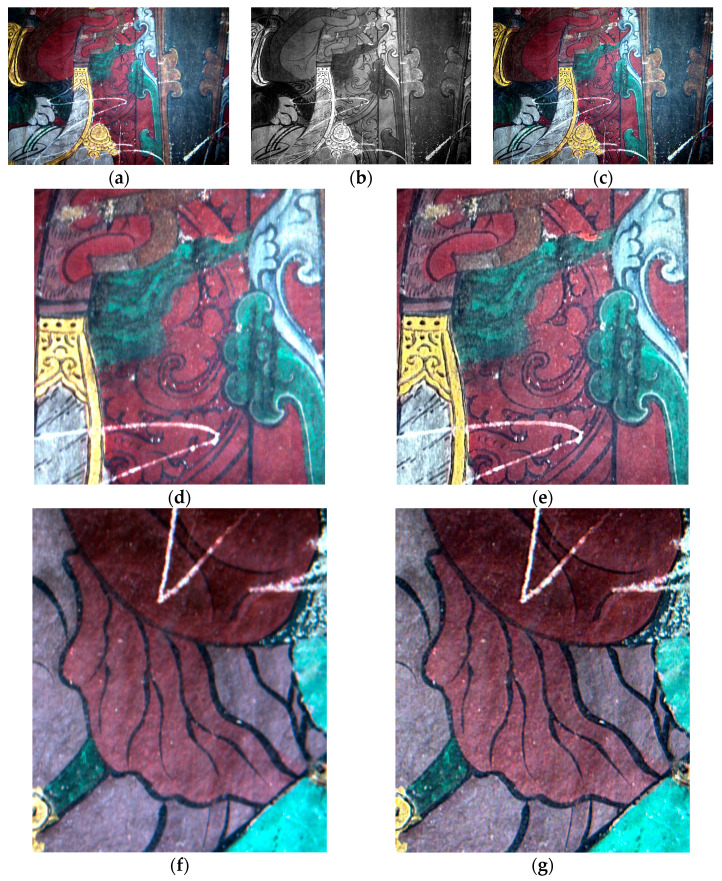
Area 1 mural enhancement results: (**a**) Area 1 true colour image; (**b**) region 1 first principal component filtering; (**c**) results of linear information enhancement; (**d**–**g**) the results of local magnification before and after enhancement of study areas 1 and 2.

**Figure 5 sensors-22-09780-f005:**
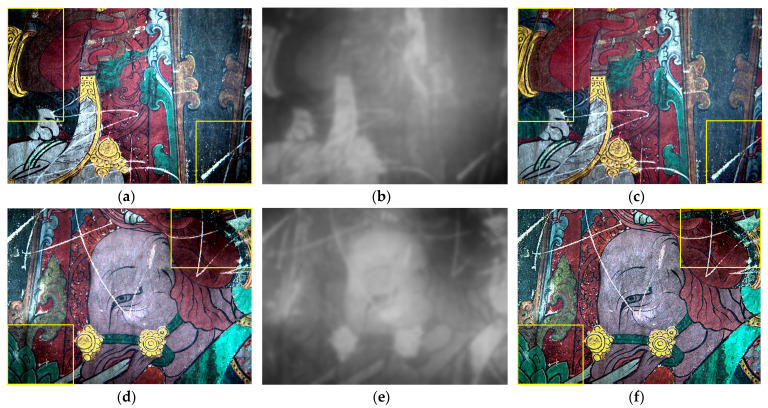
Local darkness of the mural enhancement results: (**a**) Area 1 after linear enhancement, (**b**) Area 1 light component extraction, and (**c**) Area 1 dark enhancement result; (**d**) Area 2 after linear enhancement, (**e**) Area 2 light component extraction, and (**f**) Area 2 dark enhancement result. (The yellow box is the more obvious area of change).

**Figure 6 sensors-22-09780-f006:**
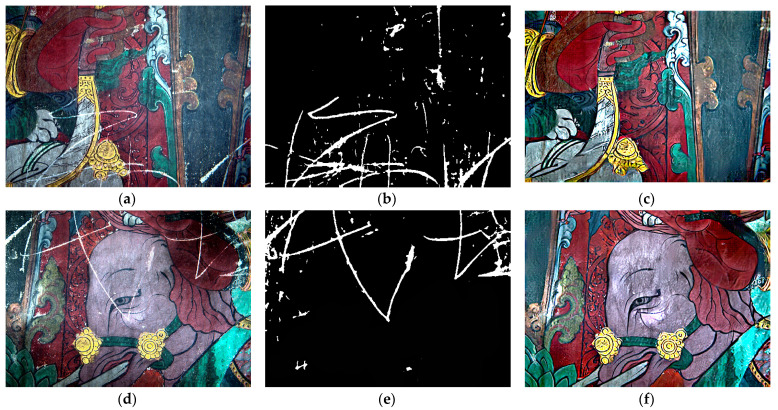
Scratch extraction and repair results: (**a**) Area 1 enhancement result, (**b**) Area 1 scratch extraction result, and (**c**) Area 1 restoration result; (**d**) Area 2 enhancement result, (**e**) Area 2 scratch extraction result, and (**f**) Area 2 restoration result.

**Figure 7 sensors-22-09780-f007:**
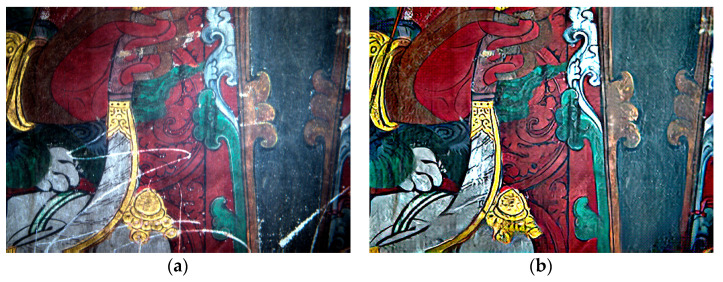
Visual comparisons between murals before and after restoration: (**a**) original and (**b**) restored maps of Area 1; (**c**) original and (**d**) restored maps of Area 2.

**Figure 8 sensors-22-09780-f008:**
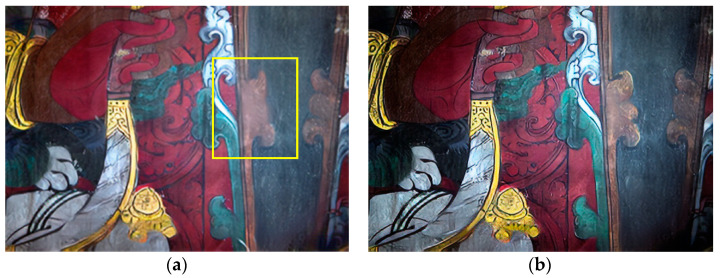
Area 1: (**a**) recovery effect without linear information enhancement and Butterworth high-pass filtering (The yellow box is an error recovery), (**b**) no local dark enhancement and Butterworth high-pass filtering recovery effect, (**c**) no Butterworth high-pass filtering, and (**d**) restoration with the proposed method.

**Figure 9 sensors-22-09780-f009:**
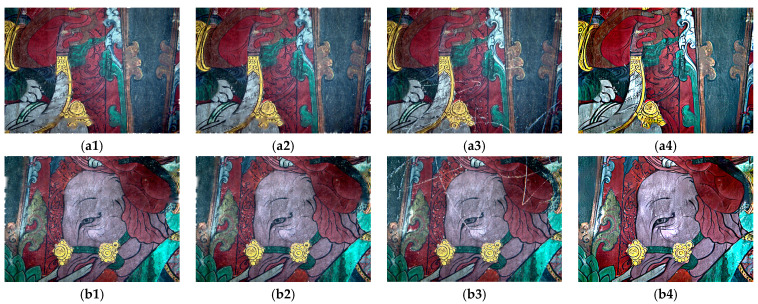
Four visual effects of different methods of repairing murals with scratches. (**a1**–**a4**,**b1**–**b4**) are the TV restoration result, CDD restoration result, Criminisi restoration result, and restoration result of the proposed method of study areas 1 and 2.

**Figure 10 sensors-22-09780-f010:**
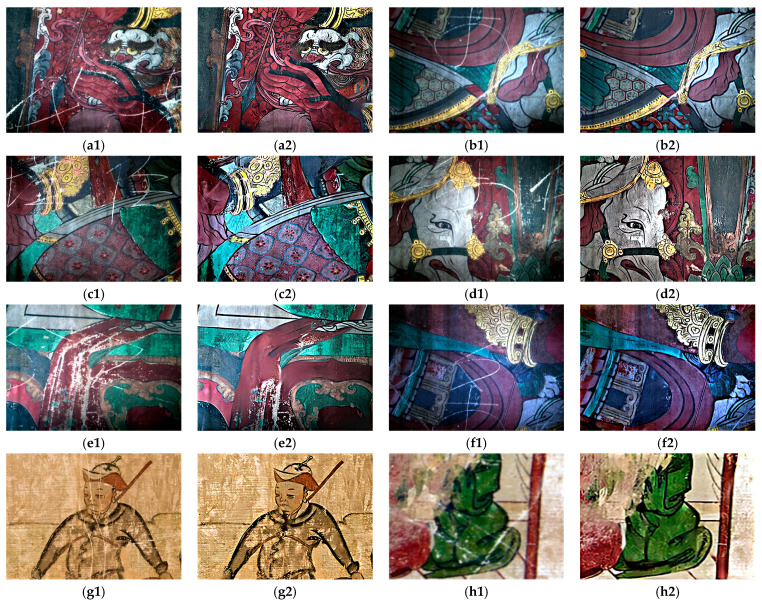
Recovery results of supplementary data. (**a1**,**b1**,**c1**,**d1**,**e1**,**f1**,**g1**,**h1**) are original images; (**a2**,**b2**,**c2**,**d2**,**e2**,**f2**,**g2**,**h2**) are the restoration results obtained using the proposed method.

**Table 1 sensors-22-09780-t001:** Experimental environment.

Environment	Parameters
Systems	Windows 10 (Microsoft, Redmond, WA, USA)
GPU	NVIDIA RTX 2080 (NVIDIA, Santa Clara, CA, USA)
CPU	i7-9700 k, CPU @3.60 GHz (8 CPUs) (Intel, Santa Clara, CA, USA)
RAM	16 GB

**Table 2 sensors-22-09780-t002:** Objective evaluations of different step combinations.

Study Area	Evaluation Indicators	No Linear Information Enhancement and Butterworth High-Pass Filtering	No Local Dark Enhancement Recovery Effect and Butterworth High-Pass Filtering	No Butterworth High-Pass Filtering	Complete Method
Area 1	Average gradient	10.1506	10.8247	14.1354	28.0683
	Edge strength	96.8508	99.6460	131.4485	168.5789
	Space frequency	23.3682	37.0902	53.5409	56.3064
Area 2	Average gradient	7.9620	10.5215	17.4626	30.0952
	Edge strength	75.4469	94.9784	157.6848	179.4221
	Space frequency	16.6812	43.2561	44.5785	48.2505

**Table 3 sensors-22-09780-t003:** The average scoring results of the subjective evaluation of the repair effect.

Method	Average Score of Area 1	Average Score of Area 2
TV	7.01	6.89
CDD	7.54	7.27
Criminisi	5.23	5.03
Proposed Method	9.12	9.07

## Data Availability

Not applicable.

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
