# Peer review of "Enhancement and Restoration of Scratched Murals Based on Hyperspectral Imaging—A Case Study of Murals in the Baoguang Hall of Qutan Temple, Qinghai, China"

_sensors, 2022, doi:10.3390/s22249780_

Round 1

Reviewer 1 Report

The article is very interesting and I have no complaints about the method used. However, the described methods are presented without evidence and supporting information. As readers, we read the proposed processing procedure and in several pictures that this procedure works. I lack evidence that the proposed methods will work like this for other images. I propose to expand the description of the described methods with the results of the evaluation of multiple images, statistical evaluation of success, parameters of designed filters, etc.

Author Response

Thank you for your review, the article quality has been significantly improved. 
We have carefully revised it based on your comments and the changes are highlighted in red. 
We will appreciate it if you could accept our modifications! The replies was attached below. 
Best regards!

Reviewer 2 Report

Missing details about training triplet domain translation network. Details about the dataset and training procedure should be provided.

Author Response

(The authors gave the same response as above.)

Reviewer 3 Report

This study aims to enhance and restore the scratched mural hyperspectral images, and proposes to combine several steps to improve the restoration performance. Experimental results manifested that the proposed method can achieve better performance than several conventional methods.

The main concerns about this study are as the following:

1) The proposed method integrates several components together to deal with the hyperspectral mural image restoration task. It is a little confused about which part is the main contribution in this study. In addition, for the purpose of the  scratched mural restoration, what kind of specific design is used in the proposed procedure.

2) The explanation of the Triplet domain translation network is unclear. What are the synthetic data, and the true value data corresponding to the synthetic data? How to train the network? What is the loss function for network training?

3) In the result section, the author mentioned that the scratches are needed to be segmented using a U-net network? There are no any explanation about the scratch segmentation in the main section. How this part affect the final enhancement results?

4) The authors introduced the conventional methods such as The total variation (TV) model, curvature-driven diffusion (CDD) model, and Criminisi algorithm for image restoration, and used the mentioned method for comparison with the proposed method. I wonder if  all the mentioned conventional methods employed many pre-processing steps? If not , the comparison would be unfair.

Author Response

(The authors gave the same response as above.)

Round 2

Reviewer 2 Report

After improvements, it seems that every part of the research is now explained. 
For future work, maybe it can be considered which wavelengths contribute mostly in the first principal component image and if they are common for all murals, to replace hyperspectral with the multispectral camera.